# Grandparental partnership status and its effects on caring for grandchildren in Europe

**Gretchen Perry** [1]*, **Martin Daly** [2]

**1** Human Services & Social Work Department, University of Canterbury, Christchurch, New Zealand,
**2** Department of Psychology, Neuroscience & Behaviour, McMaster University, Hamilton, Ontario, Canada

* gretchen.perry@canterbury.ac.nz

## Abstract

Grandparents are important childcare providers, but grandparental relationship status matters. According to several studies, caregiving is reduced after grandparental divorce, but differential responses by grandmothers *versus* grandfathers have often been glossed over. To explore the effects of relationship status on grandparental care, we analysed data from the Survey of Health, Ageing and Retirement in Europe (SHARE) comparing four grandparental relationship statuses (original couple, widowed, divorced, and repartnered) with respect to grandmothers' and grandfathers' provision of care to their birth children's children. When proximity, kinship laterality, and grandparents' age, health, employment, and financial status were controlled, divorced grandmothers without current partners provided significantly more childcare than grandmothers who were still residing with the grandfather, those who had new partners unrelated to the grandchildren, and widows without current partners. Grandfathers exhibited a very different pattern, providing substantially less grandchild care after divorce. Grandfathers in their original partnerships provided the most grandchild care, followed by widowers, those with new partners and finally those who were divorced. Seemingly contradictory findings in prior research, including studies using SHARE data, can be explained partly by failures to distinguish divorce's effects on grandmothers *versus* grandfathers, and partly by insufficient controls for the grandmother's financial and employment statuses.

## Introduction

Grandparents are significant contributors to childcare across the gamut of human societies and family types [1–3]. When divorce disrupts intergenerational relationships, it can undermine grandparental care. In a recent study of self-report data from 18 countries participating in the *Survey of Health, Ageing and Retirement in Europe* (SHARE), Žilínčíkova and Kreidl [4] concluded that "Being divorced is clearly associated with a significant reduction in the odds of providing any grandparental childcare in the past 12 months as well as in the odds of providing intensive (at least once a week) childcare".

The analyses that led Žilínčíkova and Kreidl to this conclusion did not, however, separate grandparents according to sex, and the same has been true of several studies in the United

**Data Availability Statement:** This study uses data from SHARE Waves 1, 2, 4, 5 and 6 (DOIs: 10.6103/SHARE.w1.610, 10.6103/SHARE.w2.610, 10.6103/SHARE.w4.610, 10.6103/SHARE.w5.610, 10.6103/SHARE.w6.610). These and other SHARE

data are available to "the entire research community" (see http://www.share-project.org/).

**Funding:** The authors received no specific funding for this work.

**Competing interests:** The authors have declared that no competing interests exist.

States that reached similar conclusions [5–8]. This is an important limitation, because there are a number of reasons, both theoretical and empirical, to suspect that grandmothers and grandfathers will respond differently to divorce.

Evolutionary theorists have interpreted the prolonged post-reproductive efficacy of women as evidence that grandmothering has been a specific target of natural selection [9–12], but because men, unlike women, can continue to pursue direct reproduction into old age, there is no analogous basis for suggesting the same about grandfathering. Indeed, having living grandmothers and having contact with them have been shown to be associated with child survival and wellbeing, whereas similar benefits of contact with grandfathers are weak to non-existent [13].

There are additional empirical reasons to be skeptical of claims that divorce affects grandmothers and grandfathers similarly. In analyses of data from SHARE's first wave, Hank and Buber [14], *did* distinguish between grandmothers and grandfathers, and found a significant negative effect of divorce only in the latter. Similarly, Danielsbacka and Tanskanen [15] have reported that rates of grandmaternal care in Finland were high and little affected by marital status, whereas grandpaternal care declined dramatically after divorce.

Further reasons to be skeptical of a blanket generalisation that divorce leads to a reduction in grandchild care involve the moderating effects of other variables. In one US study [16], for example, reduced grandchild care apparently characterized divorcées who had taken new partners, but not those who remained single. The same was true for grandparents generally (sexes not separated) in another US study [6], with the added wrinkle that divorce led to a significant reduction in contact with agnatic grandchildren (the children of sons) and a nonsignificant *increase* in contact with uterine grandchildren (the children of daughters). The generalization that divorced grandparents play reduced roles in grandchild care is further challenged by evidence that single grandmothers, including divorcées, play an exceptionally *large* role in the primary or custodial care of their grandchildren when parents cannot fulfill that role [17–19]. Clearly, the issue warrants further study.

Several variables are potential confounds in an analysis of the "effects" of partnership status. In the SHARE data set, as in the United States [20], grandmothers without partners are substantially more often impoverished and engaged in full-time employment than married grandmothers of the same age, and their health is substantially worse, on average, as we shall see in the analyses to follow. Widows are furthermore older, on average, than other partnership status groups. These factors may compromise a grandmother's ability to provide care, and although most investigators have controlled for grandparental age and health status, they have not always controlled for employment or financial status and no published study of which we are aware has yet controlled simultaneously for both.

Might it even be the case that divorced and widowed grandmothers actually provide *more* care than those who live with partners, once these confounds are suitably controlled? One reason for suspecting that this may be so is that couples commonly spend a large share of their discretionary or leisure time interacting with each other, with the effect that the everyday demands of being a marriage partner can "crowd out" alternative investments of time and effort [21]. Another is that grandmothers without partners may feel a particularly acute need to reinforce familial bonds of reciprocity in anticipation of needing help themselves [22]. These considerations apply with lesser force to grandfathers, because the situations and incentives confronting the two sexes are not identical. Divorced men tend not to be in such dire financial straits as their female counterparts [23, 24], and they are much more often estranged from their adult children [6, 25, 26] which is only partly a consequence of the fact that children who were still dependent when their parents divorced are relatively likely to have been in the sole or primary custody of their mother rather than their father. Moreover, although paternal

involvement in child care varies greatly across cultures and cohorts, there is a large average sex difference, with childcare everywhere consuming a larger proportion of the time and energy budgets of women than of men, whose social efforts are directed elsewhere [27].

The analyses in this paper are restricted to care of the children of a focal grandparent's birth children. The motivation for caring for a stepchild's children apparently differs from that underpinning care of one's own grandchildren [28–30], and we have analyzed such care in the SHARE data elsewhere [31].

The facts and interpretations reviewed above inspire a series of hypotheses.

*Hypothesis 1*. When confounds including financial and employment status are controlled, being divorced with no current partner will be associated with elevated levels of grandchild care by grandmothers and reduced levels among grandfathers.

*Hypothesis 2*. When confounds including financial and employment status are controlled, being widowed with no current partner will be associated with elevated levels of care by grandmothers and reduced levels of care by grandfathers.

*Hypothesis 2a*. The reduction in care by widowed grandfathers will not be as extreme as among divorced grandfathers.

The rationale for hypothesis 2a is that widowed grandfathers are less likely to be alienated from their adult children than divorced grandfathers.

*Hypothesis 3*. Grandparents who have new partners will provide less grandchild care than those who still reside with the original partner (the other grandparent).

*Hypothesis 3a*. This will be true among grandparents of both sexes, but especially among grandfathers.

The rationale for hypothesis 3a derives partly from evidence that remarried men often "move on" from their first marriages and devote their attention to investment in the new partnership [28, 29], and partly from the consideration that grandfathers' caregiving is often subsidiary to their partners' caregiving [32–34]. This point is particularly salient to the SHARE data where the question "Have you cared for your grandchildren in the past year?" (see below) does not specify caregiving acts, and is typically answered the same by both members of the couple [35].

*Hypothesis 4*. Grandparents who have new partners will provide less care than widowed or divorced grandparents who have no current partner.

## The Survey of Health, Ageing and Retirement in Europe (SHARE)

SHARE was launched in 2004 in eleven European countries plus Israel [36]. The Israeli data are excluded from this report's analyses of grandparental care in Europe. As of 2017, six waves had been completed at roughly two-year intervals. The survey's target population is persons over 50 years of age who speak an official language of the country where they reside and who are not living "in an institution such as a prison", plus their partners. In most countries, persons residing in institutions for the elderly were included in the sampling frame, but in Austria, France, Italy, and Switzerland, they were not [37].

One series of questions asked in all waves except Wave 3 addressed the frequency and nature of respondents' contacts with their children and grandchildren. In Waves 1 and 2, these questions were asked of all grandparents in contacted households, but only with reference to the first four children that they named. In subsequent waves, the same questions were asked of

only one member of a couple (usually the woman), and now with reference to up to 20 children. Analyses in this report are based on data from Waves 1, 2, 4, 5 and 6, and on responses concerning only the first four children named by an interviewee.

For each wave since the first, SHARE has recruited both new and prior interviewees. The initial and subsequent interviews of a recontacted respondent are almost identical, but there is one crucial difference. First-time participants who affirm that they have grandchildren are then asked "During the last twelve months, have you regularly or occasionally looked after your grandchild(ren) without the presence of the parents?" Participants who answer "yes" are next asked for which of their children they provided grandchild care, and for each child named, "On average, how often did you look after the child(ren) of [child name] in the last twelve months? Was it. . . (1) almost daily, (2) almost every week, (3) almost every month, or (4) less often?" In re-interviews, instead of "In the last twelve months. . .", the initial question begins "During the time since the last interview. . .", thereby asking respondents to recall grandchild care over a longer and highly variable period, ranging from about two years to more than a decade. Analyses presented here are therefore restricted to participants' first interviews.

In the first Wave, 27,984 interviews were conducted in Europe in 2004–2005. In Wave 2, another 34,727 interviews were conducted in 2006–2007, of which 14,156 (41%) were first-time interviews, and the number of participating European countries was increased to fourteen. Wave 4 added another four countries and 58,184 interviews, almost all in 2011; 36,958 (64%) were first-time interviews. Wave 5 added one more country and 63,646 interviews in 2013; 21,075 (33%) were first-time interviews. Finally, Wave 6 added a twentieth country and 66,196 interviews in 2015; 11,004 (17%) were first-time interviews. Once Wave 6 had been completed, interviews including questions about grandchild care had been administered 250,737 times, to a total of 111,177 individual Europeans: 61,948 women and 49,229 men.

## Methods

### Inclusion criteria

Criteria for eligibility for this study are that the respondent (1) was a first-time interviewee, (2) was at least 50 years old, (3) could be categorized as either having a current partner or, if not, as widowed or divorced, and (4) had at least one grandchild under the age of 13 years, whose parent was both one of the first four children listed for that respondent and the "natural" offspring (SHARE's terminology) of the respondent. The resultant data set of eligible respondents includes 19,924 women with 28,733 eligible grandchild sibling groups (henceforth called "sib sets") and 15,021 men with 21,693 eligible sib sets. It is these sib sets (50,426 in total) that are the basic units of analysis.

The reason for treating the sib set as the unit of analysis, and not the individual grandchild, is that within sib sets, the SHARE interview does not determine which particular children were actually cared for. Moreover, rather than recording the age of each grandchild, interviewers asked only the year of birth of the youngest in each sib set. Thus, although one might wish to analyze the care of children under a certain age, the best attainable approximation is to select sib sets in which the youngest met that age criterion. The age criterion chosen here was that the youngest grandchild in a sibling group was 12 years of age or younger (i.e. had not yet attained the 13th birthday), in order to restrict the cases to minor children still in need of supervision.

Inclusion criterion # 3, which limits eligibility to respondents who had either a current partner or a marital status of "widowed" or "divorced", excludes 1400 otherwise eligible sib sets whose status is ambiguous for present purposes. It is unclear, for example, how to interpret the

fact that about 1% of respondents who were recorded as "married, living with partner" were nevertheless also coded as *not* having a coresiding partner, nor can one tell whether a "never married" grandparent with no current partner formerly cohabited with the other grandparent, and if so, whether that relationship ended in a manner analogous to divorce or to widowhood.

## Dependent variables

**Grandchild care.**   Grandparental care is operationalized in two ways in the analyses reported here. The first, "Ever Care" is binary: Did the respondent affirm that she or he had cared for a particular sib set of grandchildren, in the absence of the parents, within the past year? This is the outcome variable that provided the basis for Žilínčíkova and Kreidl's conclusion that "Being divorced is clearly associated with a significant reduction in the odds of providing any grandparental childcare in the past 12 months" [4: abstract]. The second dependent variable, "Level of care" is a five-level ordinal variable ranging from (1) "Never" to (5) "Almost every day".

## Independent variables

**Grandparent's partnership status.**   Four partnership statuses are distinguished: still partnered with the other grandparent; divorced with no partner; widowed with no partner; and living with a new partner who is thus a stepparent to the grandchildren's parent.

SHARE respondents are asked to list their children, including stepchildren, adoptees, and foster children. They are then asked whether all those named are the "natural children" of the respondent and his or her partner, and if not, the particular relationships are worked out, child by child. These questions provide the basis for distinguishing between respondents whose current partners are "the other grandparent" or a "new partner". It should be noted, however, that although SHARE draws distinctions between "natural", adoptive and foster relationships with regard to respondents' relationships to their children, it does not do likewise for the next generation. Thus, the "grandchildren" for whom grandparental care is analyzed include an unknowable number who are not the respondent's lineal descendants but the adoptive, foster, or stepchildren of the respondent's birth children.

**Laterality.**   Children of a respondent's son were coded (1) and those of a daughter (2). A "matrilateral" or "uterine" bias, whereby daughters' children (*uterine* grandchildren) are cared for more than those of sons (*agnatic* grandchildren), is cross-culturally robust [for review, see 38], and this bias has already been shown to be strong and widespread in the SHARE data that are the basis for the present report [31, 39]. To the degree that this bias derives from emotional closeness between mothers and daughters, we may expect it to be stronger in grandmothers than in grandfathers, but there are theoretical as well as empirical grounds for expecting it to be present in both sexes [40].

**Proximity.**   Respondents were asked whether each of their children resided (1) 'in the same household'; (2) 'in the same building'; (3) 'less than 1 kilometre away'; (4) 'between 1 and 5 kilometres away'; (5) 'between 5 and 25 kilometres away'; (6) 'between 25 and 100 kilometres away'; (7) 'between 100 and 500 kilometres away'; (8) 'more than 500 kilometres away'; or (9) 'more than 500 kilometres away in another country'. We collapsed options (8) and (9) into a single category, and the resultant 8-point scale is our measure of proximity, reverse-coded so that higher values indicate closer proximity.

It is important to note that SHARE interviews do not elicit the distance between a respondent's residence and that of her grandchild, but only that between the respondent and her own child (the grandchild's parent), which serves as our proxy measure of respondent-grandchild proximity. Because our focus is on the care of children 12 years of age or younger, using this

measure is unlikely to have produced error due to children having become independent. However, unknown numbers of grandchildren will have been in the custody either of the grandparents themselves or of an estranged partner of the respondent's child.

**Age.**   A grandparent's age in years at interview was coded as an integer.

**Health status.**   We use a single-item measure of self-rated grandparental health: 'Would you say your health is (1) very good, (2) good, (3) fair, (4) bad, or (5) very bad?', reverse-coded to make good health the positive end of the scale. This widely used measure has good validity as a predictor of mortality, even when other health measures are controlled [41], and has recently been shown to be a valid indicator of physical and mental health in 19 European countries, including 15 of the 20 contributing data here [42].

**Economic status measures.**   Whether the interviewee was employed full time is coded as a binary (yes/no) variable. The SHARE interview also includes many detailed questions about income and assets, from which multiple estimates of both the total income and net worth of each respondent's household are derived. We used the first such estimate, and computed "adjusted household income" by dividing total household income by the square root of the number of household members in order to compensate for the lesser per capita costs in larger households [e.g., 43]. We then converted this measure and household net worth, both of which are highly skewed and cannot be assumed to be comparable over the financially turbulent 13-year period between Wave 1 and Wave 6, to wave-specific quintiles. A final, subjective measure of financial ease / distress was provided by this question: "Thinking of your household's total monthly income, would you say that your household is able to make ends meet (1) with great difficulty, (2) with some difficulty, (3) fairly easily, or (4) easily?" The bivariate correlations among the latter three measures of economic circumstances ranged from 0.4 to 0.5.

**Number of grandchild sets.**   Because grandparents must allocate finite time and efforts, we include the number of a respondent's children who themselves had children as a potential predictor of the care of particular grandchildren. For this purpose, we do not restrict the potential "competitors" for grandparental attention to grandchildren under the age of 13 years.

## Analytical methods

The unit of analysis is the grandchild set, for the reasons outlined above. The 50,426 eligible grandchild sets represent an average of 1.44 per eligible respondent. Following the recommendations of Clarke [44] and McNeish [45], who have shown that using clustered regression techniques with a great many small clusters inflates group-level variance estimates, all reported regressions were conducted without clustering. As a robustness check, however, we repeated the analyses with cases clustered within grandmothers, using Stata's "robust cluster" procedure; in no case did the clustered and unclustered analyses produce results that differed substantially or with respect to statistical significance.

Besides asking whether a respondent resides with a partner, SHARE separately records marital status in six categories: "married, living with partner"; "married, not living with partner"; "in a registered partnership"; "never married"; "widowed"; "divorced". Surprisingly, all 12 combinations of the two variables occur. We assume that "divorced", "widowed" and "never married" persons with current partners were cohabiting, and because cohabitation is now normative in most of Europe and its differences from registered marriage have largely disappeared [46], we treat the distinctions among marriages, cohabitation, and "registered partnerships" (a status recorded in 19 of the 20 countries) as moot.

The incidence of providing care was compared across relationship statuses and other variables by means of logistic regressions (using the "logistic" and "ologit" commands in Stata 13.1). Effects are reported as Odds Ratios for analyses of the predictors of "Ever Care" and as

regression coefficients for analyses of the predictors of "Level of care". Rates of caregiving are reported both as raw (actual) rates and as "predicted" probabilities of caregiving derived from logistic regressions incorporating interaction terms. All reported p values are two-tailed.

## Results

Eight groups of grandparents are distinguished in the analyses to follow, defined on the basis of grandparent sex and the four partnership statuses. Table 1 presents descriptive statistics for these eight groups. Analyses of variance, which are presented in S1 File, confirmed that there were highly significant between-group differences in relation to grandparent sex, partnership status, and/or the interaction between the two in every variable in Table 1. The differences in rates of caring might thus, in principle, be due to confounding with other variables in the table, and multivariate analyses controlling for those other variables were undertaken to dispose of the possibility that apparent "effects" of partnership status might be artifacts of these confoundings.

The descriptive statistics in Table 1 show that grandfathers exhibited much greater differences in rates of grandchild care as a function of partnership status than was the case among grandmothers. To assess whether partnership status and other variables affect caregiving of grandmothers and grandfathers in similar ways, Table 2 presents the results of a series of logistic regression models incorporating various predictors of the binary outcome variable "Ever care". "Divorced," "Widowed," and "New Partner" are dummy (yes / no) variables, whose reported effects are the ratios of their odds of providing care divided by the odds for those still residing with the other grandparent. Table 3 presents the results of parallel ordered logistic models predicting the 5-level ordinal outcome variable "Level of care".

### Hypothesis 1

When confounds including financial and employment status are controlled, being divorced with no current partner will be associated with elevated levels of grandchild care by grandmothers and reduced levels among grandfathers.

This hypothesis was confirmed. In both Tables 2 and 3, in the absence of controls for economic circumstances (Model 1), divorce is associated with a major reduction in care by grandfathers, and a lesser reduction in care by grandmothers. This is consistent with most published findings. In Model 2, however, with the novel addition of simultaneous controls for both full-time employment and financial status, the apparent impact of divorce on grandmaternal care is reversed, becoming significantly positive, whereas the results for grandfathers are relatively unaffected. In short, in support of Hypothesis 1, care by grandmothers after divorce is actually elevated, not suppressed, net of the effects of all other predictors in Model 2.

Other potential predictors of grandparental care retain highly significant effects in Model 2 in both Tables 2 and 3. Proximity, good health, being relatively young, and being financially well off apparently facilitate care by grandmothers and grandfathers to roughly similar degrees, and full-time employment has roughly similar negative effects on case by grandparents of either sex. However, it is only in grandmothers that there is any evidence that grandchild sibling sets "compete" for the respondent's time. Preferential care of daughters' children over those of sons is strong in both sexes, especially grandmothers.

### Hypothesis 2

When confounds including financial and employment status are controlled, being widowed with no current partner will be associated with elevated levels of grandchild care by grandmothers and reduced levels among grandfathers.

**Table 1. Descriptive statistics.**

| | n | Percent provided any care | Age mean ± SD | Proximity (8-pt scale) mean ± SD | Health (5-pt scale) mean ± SD |
|---|---|---|---|---|---|
| **Grandmothers** | | | | | |
| with grandfather | 18945 | 58.2 | 62.6 ± 7.2 | 4.29 ± 1.7 | 2.91 ± 1.1 |
| divorced, alone | 2662 | 56.5 | 61.4 ± 7.0 | 4.03 ± 1.7 | 2.80 ± 1.1 |
| widowed, alone | 5357 | 46.4 | 68.4 ± 8.1 | 4.42 ± 1.8 | 2.67 ± 1.1 |
| new partner | 1769 | 52.7 | 59.5 ± 6.1 | 3.72 ± 1.6 | 2.93 ± 1.1 |
| **Grandfathers** | | | | | |
| with grandmother | 18091 | 49.8 | 65.6 ± 7.7 | 4.26 ± 1.7 | 2.93 ± 1.1 |
| divorced, alone | 1095 | 24.0 | 62.5 ± 7.2 | 3.77 ± 1.6 | 2.97 ± 1.2 |
| widowed, alone | 1141 | 32.3 | 70.9 ± 8.2 | 4.37 ± 1.8 | 2.75 ± 1.1 |
| new partner | 1366 | 27.8 | 63.2 ± 7.3 | 3.39 ± 1.5 | 3.07 ± 1.1 |

| | Percent working full time | Financial ease (4-pt scale) mean ± SD | Household net worth (1000s of Euros) median / mean | Adjusted household income (1000s of Euros) mean ± SD |
|---|---|---|---|---|
| **Grandmothers** | | | | |
| with grandfather | 23.4 | 2.77 ± 0.98 | 147.0 / 258.1 | 26.1 ± 0.28 |
| divorced, alone | 33.1 | 2.26 ± 1.01 | 28.0 / 100.8 | 15.6 ± 0.40 |
| widowed, alone | 9.9 | 2.44 ± 0.99 | 56.1 / 147.7 | 14.8 ± 0.33 |
| new partner | 39.0 | 2.88 ± 0.86 | 101.0 / 239.1 | 30.1 ± 1.03 |
| **Grandfathers** | | | | |
| with grandmother | 23.5 | 2.79 ± 0.99 | 150.2 / 278.9 | 28.1 ± 0.36 |
| divorced, alone | 31.1 | 2.65 ± 1.06 | 51.1 / 155.7 | 20.9 ± 0.97 |
| widowed, alone | 9.9 | 2.82 ± 1.03 | 100.1 / 235.1 | 20.8 ± 0.85 |
| new partner | 38.1 | 3.00 ± 0.95 | 138.1 / 294.7 | 35.9 ± 1.25 |

The enumerated cases are grandchild "sib sets" as described in the Methods section.

**Table 2. Results of a series of logistic regressions predicting whether interviewees had provided care for grand-children ("Ever care") within the year prior to interview.** (Country of residence differences were controlled by the inclusion of country-specific dummy variables, the results for which are not shown).

| | Grandmothers | | | | | |
|---|---|---|---|---|---|---|
| Model | 1 | | | 2 | | |
| | O.R. | 95% CI | p | O.R. | 95% CI | p |
| Grandparents still together: *reference* | | | | | | |
| Divorced, no partner | 0.92 | 0.84–1.01 | .066 | 1.21 | 1.11–1.33 | .000 |
| Widowed, no partner | 0.84 | 0.78–0.92 | .000 | 1.02 | 0.95–1.10 | .619 |
| New unrelated partner | 0.73 | 0.65–0.81 | .000 | 0.74 | 0.67–0.83 | .000 |
| Proximity (8-point scale) | 1.38 | 1.36–1.40 | .000 | 1.40 | 1.37–1.42 | .000 |
| Daughter's child (reference: son's) | 1.53 | 1.46–1.61 | .000 | 1.54 | 1.46–1.62 | .000 |
| Age (years) | 0.96 | 0.96–0.97 | .000 | 0.96 | 0.95–0.96 | .000 |
| Health (5-point scale) | 1.25 | 1.22–1.28 | .000 | 1.17 | 1.14–1.20 | .000 |
| N of grandchild sib sets | 0.88 | 0.86–0.90 | .000 | 0.89 | 0.87–0.91 | .000 |
| Full time employment | | | | 0.87 | 0.80–0.93 | .000 |
| Financial ease (4-point scale) | | | | 1.11 | 1.07–1.14 | .000 |
| Household income quintile | | | | 1.15 | 1.12–1.18 | .000 |
| Net household worth quintile | | | | 1.15 | 1.13–1.18 | .000 |
| N cases | 28604 | | | 28558 | | |
| Nagelkerke pseudo-$R^2$ | .172 | | | .191 | | |
| AIC / N | 1.239 | | | 1.222 | | |
| | Grandfathers | | | | | |
| Model | 1 | | | 2 | | |
| | O.R. | 95% CI | p | O.R. | 95% CI | p |
| Grandparents still together: *reference* | | | | | | |
| Divorced, no partner | 0.28 | 0.24–0.32 | .000 | 0.33 | 0.28–0.38 | .000 |
| Widowed, no partner | 0.50 | 0.44–0.58 | .000 | 0.55 | 0.48–0.63 | .000 |
| New unrelated partner | 0.37 | 0.33–0.42 | .000 | 0.37 | 0.32–0.42 | .000 |
| Proximity (8-point scale) | 1.33 | 1.30–1.35 | .000 | 1.35 | 1.33–1.38 | .000 |
| Daughter's child (reference: son's) | 1.44 | 1.36–1.52 | .000 | 1.45 | 1.37–1.53 | .000 |
| Age (years) | 0.98 | 0.97–0.98 | .000 | 0.97 | 0.96–0.97 | .000 |
| Health (5-point scale) | 1.16 | 1.13–1.19 | .000 | 1.10 | 1.07–1.14 | .000 |
| N of grandchild sib sets | 1.02 | 0.99–1.04 | .279 | 1.01 | 0.99–1.04 | .308 |
| Full time employment | | | | 0.76 | 0.70–0.82 | .000 |
| Financial ease (4-point scale) | | | | 1.14 | 1.10–1.19 | .000 |
| Household income quintile | | | | 1.13 | 1.10–1.16 | .000 |
| Net household worth quintile | | | | 1.15 | 1.12–1.18 | .000 |
| N cases | 21605 | | | 21576 | | |
| Nagelkerke pseudo-$R^2$ | .147 | | | .171 | | |
| AIC / N | 1.266 | | | 1.247 | | |

Tables 2 and 3 both show that this hypothesis was upheld only for grandfathers. In grand-mothers, adding these economic controls abolished the apparent negative effect of widowhood but did not transform it into a positive effect.

## Hypothesis 2a

The reduction in care by widowed grandfathers will not be as extreme as among divorced grandfathers.

**Table 3. Results of a series of ordered logistic regressions predicting "Level of care" from the same sets of predictors as in Table 2.**

| | Grandmothers | | | |
|---|---|---|---|---|
| Model | 1 | | 2 | |
| | Coefficient (SE) | p | Coefficient (SE) | p |
| Grandparents still together: *reference* | | | | |
| Divorced, no partner | - .112 (.039) | .004 | .125 (.041) | .002 |
| Widowed, no partner | - .172 (.033) | .000 | - .009 (.034) | .783 |
| New unrelated partner | - .340 (.048) | .000 | - .309 (.048) | .000 |
| Proximity (8-point scale) | .450 (.007) | .000 | .460 (.008) | .000 |
| Daughter's child (reference: son's) | .461 (.023) | .000 | .465 (.023) | .000 |
| Age (years) | - .032 (.002) | .000 | - .042 (.002) | .000 |
| Health (5-point scale) | .190 (.011) | .000 | .148 (.012) | .000 |
| N of grandchild sib sets | - .136 (.011) | .000 | - .131 (.011) | .000 |
| Full time employment | | | - .301 (.031) | .000 |
| Financial ease (4-point scale) | | | .079 (.014) | .000 |
| Household income quintile | | | .106 (.011) | .000 |
| Net household worth quintile | | | .117 (.010) | .000 |
| N cases | 28604 | | 28558 | |
| Nagelkerke pseudo-$R^2$ | .201 | | .214 | |
| AIC / N | 2.675 | | 2.660 | |
| | Grandfathers | | | |
| Model | 1 | | 2 | |
| | Coefficient (SE) | p | Coefficient (SE) | p |
| Grandparents still together: *reference* | | | | |
| Divorced, no partner | - 1.282 (.074) | .000 | - 1.132 (.075) | .000 |
| Widowed, no partner | - .745 (.067) | .000 | - .664 (.067) | .000 |
| New unrelated partner | - .971 (.063) | .000 | - .972 (.063) | .000 |
| Proximity (8-point scale) | .379 (.009) | .000 | .396 (.009) | .000 |
| Daughter's child (reference: son's) | .393 (.027) | .000 | .396 (.027) | .000 |
| Age (years) | - .018 (.002) | .000 | - .032 (.002) | .000 |
| Health (5-point scale) | .128 (.013) | .000 | .091 (.014) | .000 |
| N of grandchild sib sets | - .008 (.013) | .531 | - .011 (.013) | .417 |
| Full time employment | | | - .372 (.038) | .000 |
| Financial ease (4-point scale) | | | .119 (.017) | .000 |
| Household income quintile | | | .104 (.013) | .000 |
| Net household worth quintile | | | .122 (.012) | .000 |
| N cases | 21605 | | 21576 | |
| Nagelkerke pseudo-$R^2$ | .160 | | .178 | |
| AIC / N | 2.475 | | 2.457 | |

This hypothesis is supported by the data in Model 2 in both Tables. Confirmation that this contrast is significant was provided by running Model 2 (analysis not shown) with only two partnership status groups, namely divorced and widowed grandfathers; for the "Ever care" analysis, the odds ratio for care by divorced grandfathers relative to that by widowed grandfathers was 0.59, p = .000.

## Hypothesis 3

Grandparents who have new partners will provide less grandchild care than those who still reside with the original partner (the other grandparent).

This hypothesis is supported for both grandmothers and grandfathers by the highly significant negative effects on care associated with "New unrelated partner" in all models in Tables 2 and 3.

### Hypothesis 3a

This will be true among grandparents of both sexes, but especially among grandfathers.

The results in Tables 2 and 3 also support this hypothesis. The significance of the contrast between reduced care by grandmothers and grandfathers was tested by additional regressions (analyses not shown) in which an interaction between partnership status and sex of grandparent was included; for "Ever care", the interaction was shown to be significant in the predicted direction by a likelihood ratio test ($x^2_{\text{1df}} = 82.3$, p = .000).

### Hypothesis 4

Grandparents who have new partners will provide less care than widowed or divorced grandparents who have no current partner.

The results in Tables 2 and 3 support this hypothesis only in the case of grandmothers. The new-partners *versus* no-partner contrasts were confirmed by running Model 2 without the grandparents still together group and with the repartnered grandmothers as the reference group (analyses not shown). For "Ever care", the odds ratio for care by widows relative to those with new partners was 1.34 (p = .000), and for divorcees relative to those with new partners it was 1.58 (p = .000). For "Level of care", the coefficient for widows relative to repartnered women was 0.422 ± .063 (SE), p = .000, and for divorcées it was 0.471 ± .062, p = .000.

For grandfathers, however, the hypothesis was supported only relative to the widowed group, with divorced grandfathers without current partners providing the least care and grandfathers in new partnerships being intermediate between the divorced and widowed grandfathers. When the same analyses were completed for grandfathers as for grandmothers (analyses not shown), widowers indeed provided significantly more care than repartnered grandfathers (OR = 1.36, p = .000; coefficient = 0.409 ± .107 (SE), p = .000), whereas divorced grandfathers did not differ significantly from repartnered grandfathers (OR = .86, p > .1; coefficient = -0.130 ± .102 (SE), p > .1).

**Additional robustness checks.** Table 4 presents the results of three additional regressions testing the robustness of the effects of divorce and widowhood on "Ever care" against potential challenges. In Model 3, the reference group consisting of the progeny of a "natural" child of both members of a coresiding couple was restricted to respondents whose marital status was recorded as "married, living with partner". The rationale for this control comparison is that "divorced" and "widowed" are labels that may not have captured former cohabiting relationships, and insofar as registered marriages and cohabiting couples differ, the apparent effects of divorce and widowhood could, in principle, have been artifacts of comparing incommensurate groups. The Model 3 results obviate this concern: only 2% of the couples group were eliminated on this basis, and the results for all predictors are virtually identical to those found in Model 2.

Model 4 in Table 4 assesses the effect of removing employment status from the list of control variables. Divorced grandmothers are more likely to have full-time jobs than their married counterparts, and if employment reduces the time available for grandchild care, controlling for it might remove an important mediator of a "real" reduction in care by divorcées. However, the effects of divorce and the other remaining predictors in Model 4 are virtually unchanged from those in Model 2, Table 2. In a similar vein, Model 5 assesses the effect of removing proximity as a control variable. The rationale for this analysis is that how close one lives to one's

**Table 4. Results of further logistic regressions run to assess the robustness of the effects of divorce, widowhood, and repartnering on care by grandmothers.**

| Model | 3 | | | 4 | | | 5 | | |
|---|---|---|---|---|---|---|---|---|---|
| | O.R. | 95% CI | p | O.R. | 95% CI | p | O.R. | 95% CI | p |
| Grandparents still together: *reference* | | | | | | | | | |
| Divorced, no partner | 1.20 | 1.09–1.32 | .000 | 1.20 | 1.09–1.31 | .000 | 1.11 | 1.01–1.21 | .025 |
| Widowed, no partner | 1.02 | 0.94–1.09 | .688 | 1.01 | 0.94–1.09 | .797 | 1.06 | 0.99–1.13 | .126 |
| New unrelated partner | 0.74 | 0.65–0.84 | .000 | 0.74 | 0.66–0.82 | .000 | 0.66 | 0.59–0.73 | .000 |
| Proximity (8-point scale) | 1.40 | 1.37–1.42 | .000 | 1.39 | 1.37–1.42 | .000 | | | |
| Daughter's child (reference: son's) | 1.53 | 1.46–1.62 | .000 | 1.54 | 1.46–1.62 | .000 | 1.47 | 1.40–1.55 | .000 |
| Age (years) | 0.96 | 0.95–0.96 | .000 | 0.96 | 0.96–0.96 | .000 | 0.96 | 0.95–0.96 | .000 |
| Health (5-point scale) | 1.16 | 1.13–1.19 | .000 | 1.16 | 1.13–1.19 | .000 | 1.14 | 1.11–1.17 | .000 |
| N of grandchild sib sets | 0.88 | 0.86–0.90 | .000 | 0.89 | 0.87–0.91 | .000 | 0.89 | 0.87–0.91 | .000 |
| Full time employment | 0.85 | 0.79–0.92 | .000 | | | | 0.90 | 0.84–0.97 | .004 |
| Financial ease (4-point scale) | 1.10 | 1.07–1.14 | .000 | 1.10 | 1.07–1.14 | .000 | 1.10 | 1.06–1.13 | .000 |
| Household income quintile | 1.15 | 1.12–1.18 | .000 | 1.14 | 1.11–1.17 | .000 | 1.11 | 1.09–1.14 | .000 |
| Net household worth quintile | 1.15 | 1.13–1.18 | .000 | 1.15 | 1.13–1.18 | .000 | 1.14 | 1.12–1.17 | .000 |
| N cases | 27378 | | | 28558 | | | 28687 | | |
| Nagelkerke pseudo-$R^2$ | .193 | | | .191 | | | .117 | | |
| AIC / N | 1.220 | | | 1.223 | | | 1.284 | | |

See text for explanations of the various models. Table entries are Odds Ratios. ** p < .01; *** p < .001.

children and grandchildren is, at least in part, a choice, so variability in proximity could be interpreted as an actively selected component of relative alienation from family rather than as an independent factor mitigating opportunities for contact and childcare. It is noteworthy, then, that divorced grandmothers persist in having a significantly higher incidence of care than those still married under Model 5, and that the tendency for widows to do likewise also persists. In short, the inclusions of employment status and proximity as control variables have not artifactually masked a negative effect of divorce on grandmothers' contact and involvement with their grandchildren.

Fig 1 portrays the predicted probabilities that grandparents in the different partnership statuses would provide grandchild care, derived from a logistic regression incorporating all of the predictors in Table 2's Model 2 plus a partnership-status by sex-of-grandparent interaction term. The figure reinforces the conclusion that being divorced has a large negative impact on care by grandfathers and a smaller (albeit still highly significant) positive impact on care by grandmothers. It is easy to see how analysts might perceive only a negative effect of divorce if they included main effects in their models but failed to incorporate this interaction.

## Discussion

Contradicting previous research, the analyses here indicate that divorced grandmothers in Europe provide significantly more care to their grandchildren than those still living with the children's grandfather. The results in Tables 2 and 3 and Fig 1 demonstrate this positive effect of divorce, net of the effects of several variables that are confounded with divorce, namely residential proximity to the relevant children, and the grandmother's age, health, employment, and financial situation. This positive effect of divorce has been overlooked in prior research, including studies based on the same SHARE dataset, for two reasons. Žilínčíkova and Kreidl [4] noted only an overall negative effect of divorce on grandchild care because they failed to

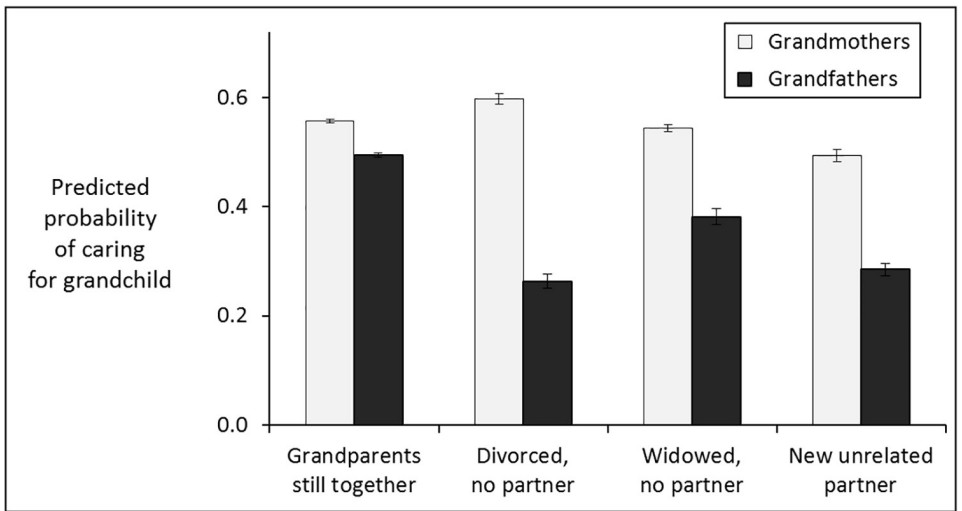

**Fig 1. Predicted probabilities (± standard errors) of providing care for grandchildren who were the progeny of the respondent's own (birth) children, based on a logistic regression incorporating all the predictors in Model 2 of Table 2.**

separate effects on grandfathers *versus* grandmothers and a large negative effect on the former overwhelmed the smaller but significant positive effect on the latter. Hank and Buber [14] observed that divorce was associated with reduced care only in grandfathers, but without any statistical controls for financial status, they failed to note that the association is positive in grandmothers. Knudsen [32] reached similar conclusions in comparisons between respondents with and without partners.

It is no surprise that divorce, which has until recently been strongly associated with maternal custody of children and reduced contact with fathers and their kin, should have a negative effect on care by grandfathers [47]. It is perhaps more surprising that it is associated with an actual increase in care by grandmothers. One possible explanation for such an effect is that a woman with no partner "consuming" her time and attention may have more free time to spend with her grandchildren. Older single women may also be especially motivated to reinforce ties with their adult children in anticipation of their own eventual dependency [22]. Many divorcées experience reduced social networks post-divorce, perhaps resulting in an increased current need for functional and material support, as well as reliable emotional connection [48–50]. These are issues deserving of further investigation.

The patterns of grandparental care and the effects of divorce that we report here are not universal. When the regression analyses in Model 2, Tables 2 and 3, are run separately for each country, for example, all 20 participating countries exhibit a negative effect of divorce on care by grandfathers, but only 13 of the 20 exhibit the positive effect in grandmothers; we have not presented these results in detail because country-specific Ns are small and confidence intervals are large. In addition to variability between countries, there is surely variability over time. Divorce rates have risen dramatically in Europe in recent decades [51], as have various forms of joint custody [52], and the stigma associated with divorce has waned [53], so we cannot expect sex differences in care and in the effects of divorce to be stable.

## Limitations

Unfortunately, the data analyzed in this paper are not suitable for disentangling effects of survey year and grandparental birth cohort from those of country differences. New countries

have been added to SHARE at each wave, and after a country's first participation, the numbers of new intervewees in any subsequent wave have been modest. Thus, for example, Wave 1 provides 48% of all cases that we analyzed from the 11 European countries that first participated then, and Wave 2 provides 63% of all cases for the three countries that joined the survey at that point. This is because SHARE prioritizes the acquisition of longitudinal data from repeat participants, and new interviewees are recruited mainly as replacements for attrition. We nevertheless confined our analyses to new participants because re-interviews inquire about grandchild care over a longer and more variable interval than the 12 months specified in the first interview. Perhaps, with suitable attention to this complication, the longitudinal aspect of the SHARE survey can be exploited to shed light on secular trends, but that is a task for future analysts. A further complication that will require attention in any such analysis is that most of the data from western Europe were collected in Waves 1 and 2, before the financial crisis of 2008–2009, whereas most of the data from eastern Europe were collected in 2010–2015. Determining the impact of the financial crisis on social change generally, and grandparental caregiving specifically, is an area for future research consideration.

The research reported here is based on self-report data, which necessarily entails some risk of invalidity due to faulty memory and/or biased self-presentation. Moreover, whether a potential respondent agrees to participate in such a survey cannot be assumed to be independent of marital history and grandchild care. The complex cross-national implementation of the SHARE survey entailed many country-specific decisions. The research teams in different countries encountered a diversity of problems in identifying and accessing the target population, and based their sampling on methods ranging from the use of a national registry through regional directories (with some regions opting out) to starting from a telephone directory; all participating countries except Austria reportedly "attempted proper probability sampling" in Wave 1, but only a minority fully achieved this objective [37].

Further limitations derive from details of what SHARE respondents were and were not asked. Our measures of caregiving are based on questions about whether respondents had cared for grandchildren "in the absence of the parents", but are silent on the specifics of such care. The fact that this question was asked only with respect to a "grandchild set" (a sibling group) limits inferences about such distinctions as caring for infants *versus* minding school-age children. Moreover, a "yes" response is not necessarily an affirmation of "hands-on" childcare, and it is plausible that a couple who have a grandchild stay overnight with them might both say "yes" on that basis alone, even if only one plays an active caregiving role. This seems a likely explanation for a strong positive contingency between grandmother and grandfather care [35]. Indeed, it is possible that grandfathers provide substantially less care than grandmothers to a similar degree regardless of partnership status, but that partnered grandfathers often affirmed caregiving when they were merely present while the grandmother provided care.

Some European grandparents are surely their grandchildren's primary caretakers, but SHARE does not ask if this is the case, and such cases cannot readily be identified, with the result that using the proximity of a grandparent's residence to her child's residence as a proxy for grandparent-grandchild proximity must have introduced error. Coall and collaborators [54] have proposed that even grandparents who look after the children "almost daily" (the maximum option) must rarely be custodial grandparents, because they seldom claim to have spent more than 10 hours a day in caregiving (a variable that we did not make use of). We question this inference, because a custodial grandparent cannot be assumed to count the hours when children were asleep, at school, out with friends, or simply out of sight, as hours spent caring for them. It is also not possible to identify those grandchildren who are in the custody of an estranged partner of the respondent's child, which will have caused further error in the

proximity measure, nor is it possible to identify grandchildren who are not the respondent's lineal descendants because they are the step-, adoptive or foster children of the respondent's birth child.

Also of relevance to the analyses in this paper is the fact that the existence and fate of former cohabitations were not determined, so which ones ended in a way analogous to divorce and which by the partner's death is unknown. The distinction between registered marriage and cohabitation is no longer of great importance in most of Europe [46], and we treat the "natural children" of both members of a coresiding couple accordingly, as a single category regardless of marital status. We were unable to define the "divorced" and "widowed" groups in an equally encompassing way, but the virtually identical results of Models 3 (Table 3) and 4 (Table 4) indicate that this is unlikely to have affected the findings.

## Conclusion

Blanket claims that divorce disrupts relationships with adult children and therefore reduces grandchild care are mistaken, at least in the case of European grandmothers. Divorcées are actually more likely to provide care than those grandmothers still married to the grandfather or widows, once relevant confounds are controlled. Grandfathers, however, are substantially less likely to provide grandchild care in any relationship status other than continued marriage to the grandmother.

## Supporting information

**S1 File.**
(DOCX)

## Acknowledgments

We thank Mirkka Danielsbacka, David Coall, Ralph Hertwig, Sonja Hilbrand, and Antti Tanskanen for collegial discussion, and further thank Mirkka and Antti for their guidance concerning the manipulation of SHARE data files.

## Author Contributions

**Conceptualization:** Gretchen Perry, Martin Daly.

**Data curation:** Gretchen Perry, Martin Daly.

**Formal analysis:** Martin Daly.

**Project administration:** Gretchen Perry.

**Validation:** Gretchen Perry.

**Visualization:** Gretchen Perry, Martin Daly.

**Writing – original draft:** Gretchen Perry, Martin Daly.

**Writing – review & editing:** Gretchen Perry, Martin Daly.

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
