## [Decision Letter · Decision Letter 0]

20 Jan 2021

PONE-D-20-38406

Grandparental partnership status and its effects on caring for grandchildren in Europe.

PLOS ONE

Dear Dr. Perry,

Thank you for submitting your manuscript to PLOS ONE. After careful consideration, we feel that it has merit but does not fully meet PLOS ONE’s publication criteria as it currently stands. Therefore, we invite you to submit a revised version of the manuscript that addresses the points raised during the review process.

The reviewer #1 had some comments and pointed out several methodological concerns, which I share. In addition to those (some might overlap), I had several comments that need to be either addressed or clarified in the revised text.

1. All the hypotheses proposed in the introduction are informative/directional hypotheses but statistical testing of these hypotheses are traditional two-sided tests. The authors may want to explore the possibility that the conclusions might change given the more exact tests (reduced type II error rate). Moreover, do your analyses actually test for the hypotheses #3 and #4 as these are about grandparents but your models (in Table 2) provide results by grandparent sex? I mean you do not seem to provide any formal tests (seeing 95% CIs for odds ratios would help) for the differences of the coefficients between grandmothers and grandfathers, nor you model any interactions to examine such questions (except hypothesis #3a).

2. It was my impression that sib set size was not accounted for in any way? I understand that multilevel modeling is probably an overkill here but could a larger sib set mean increased odds that someone in the sib set was cared for? So would it make any sense to use sib set size as a predictor?

3. I was unclear of the rationale to restrict the main analyses for binary response only, potentially hiding important variation in grandparental care? Furthermore, regarding to S1 Table 1 OLS regression may not be well suited here since the scale of the response is ordinal. Thus, ordered logistic (multinominal) model would probably have been more appropriate. Thus, I am not fully convinced that treating care as binary variable does not miss anything important here.

4. Measurement of financial status. It is unclear to me why the two measures of financial status were converted to quintiles? Was this because they were skewed? If so, why is this problematic? Regression analyses make no distributional assumptions about the independent variables. Moreover, why you think multicollinearity (which doesn’t seem too dire in this case, and might be better evaluated using variance inflation factors) would be a problem here? You are using these variables as confounders so you make no inference on them. So why bother if the SEs of these control variables are a bit off? In my opinion, just putting all three into the model is thus a better idea than using a sum score since its reliability is not too great (one probably should use coefficient omega instead of alpha, and such cut-off values are likely context-specific, see e.g. McNeish 2018) - app. 30% of its variance is measurement error, and thus you are estimating your confounder effect with error. If financial status is a true causal confounder here, you thus fail to properly account for its effect by using a proxy variable. If you choose to use your current approach, the appropriate method is to use e.g. structural equation model where you can model such measurement error in this scale.

McNeish, D. 2018 Thanks coefficient alpha, we'll take it from here. Psych. Methods 23, 412-433.

5. As commented by the reviewer #1, what’s the rationale for fitting three models e.g. in Table 2? In general, you are testing a priori defined hypotheses with confounding factors and thus there is no real need for fitting multiple models or model selection. If the goal was to show the difference in results when adding more confounders, that’s fine but then why model 1?

Some minor comments:

1. Is Stata's "robust cluster" approach a design-based method (like GEE)? I don’t understand line 273-275. Do you mean you computed so called “marginal effects”? If yes, what interactions got to do with these? Please clarify.

2. Also, please use conventional statistical terminology throughout instead of software-specific nomenclature and provide exact p-values as well as errors for the point estimates, which all are good statistical practice.

3. Country should be a natural choice as a random factor in these kinds of analyses. Why you choose not to include it into the model?

We look forward to receiving your revised manuscript.

Kind regards,

Samuli Helle

Academic Editor

PLOS ONE

Journal Requirements:

Reviewers' comments:

Reviewer's Responses to Questions

**Comments to the Author**

1. Is the manuscript technically sound, and do the data support the conclusions?

Reviewer #1: Partly

Reviewer #2: Yes

2. Has the statistical analysis been performed appropriately and rigorously? 

Reviewer #1: I Don't Know

Reviewer #2: Yes

3. Have the authors made all data underlying the findings in their manuscript fully available?

Reviewer #1: Yes

Reviewer #2: No

4. Is the manuscript presented in an intelligible fashion and written in standard English?

Reviewer #1: Yes

Reviewer #2: Yes

5. Review Comments to the Author

Reviewer #1: Interesting look at partnership status and the association with grandchild-directed caregiving. The introduction could do with tightening up – at present, it is not immediately clear what this study is addressing, nor is there a strong theoretical background. I have some concerns with the methodology, as this section is rather unclear too.

Comments:

From lines 38-46, it seems that grandmothering is the focus, but then the question on line 46 (regarding grandmothers) is followed with discussion on grandparents without separating the sexes (something you then criticise in the following paragraph). There are other points where grandmothering and grandparenting are used almost interchangeably (e.g. line 77-78).

Line 39 – to help contextualise this for the reader, give examples of what would be a high-risk family situation

Line 55 – I suggest rewording the start of the paragraph as the language here might be a little strong. This previous paper (by Žilínčíkova and Kreidl) states that they attempted to include grandparental gender but “found no (statistically) significant interactions”. Their omission of these results is unusual, certainly, but I would hesitate to call their conclusions premature.

Line 107 – what are the theoretical grounds for expecting it in both sexes? Give brief examples here

Reference 29 is listed as “manuscript under review” – this should not be in the reference list as readers will not be able to find it until it has been published. One option here is to upload this paper as a citable pre-print.

Introduction talks about uterine and agnatic grandchildren, but this is not then used in the hypotheses. Remove from here and add to methods in ‘Laterality’ section (line 214) to show why you control for laterality

These hypotheses lean heavily on the idea that divorce will result in estrangement for grandfathers – lines 90-93 “they are much more often estranged from their adult children, largely because children who were still dependent when their parents divorced are relatively likely to have been in the sole or primary custody of their mother rather than their father”. However, from the methods it does not appear that SHARE data includes information on timing of separation, so this is a dangerous a priori assumption to make. Similarly, “new partners” in hypotheses 3 and 4 does not appear to account for timing – one would expect that getting a new partner in the previous year vs ten years earlier may be different in the time they spend with each other versus biological grandchildren.

Were there cases in the SHARE data included where both grandparents in a partnership were interviewed? If so, were these both included? These would be non-independent, and the current analytical structure does not appear to control for this if it has arisen as an issue.

Was any attempt made to control for the country of the respondent? The authors conclude the paper with a statement that “blanket claims...are mistaken”, yet without controlling for potential between-country differences, the claims in this paper can also be considered ‘blanket’ for Europe.

Why did you divide the financial status measure into quintiles as opposed to e.g. quartiles? The financial distress measure should be simplified too for the purposes of calculation (difficult vs easy) - self-reported “with great difficulty” and “with some difficulty” (and the same for the ‘easily’ responses) is heavily subjective. The justification in the supplementary does not really justify the arbitrary way in which this whole financial status variable was created, other than by saying the results didn’t differ for grandmaternal care. What about grandfathers, since the paper is about grandparents and not just grandmothers? Is there a reason why this comparison is not presented?

It is unclear from the text what were the final model variables, and what was the justification for having three models – table 2 shows full time employment status were included, but it does not clearly mention this in the methods. On a related note, were correlations between variables checked? In particular, full-time employment and financial status may be correlated, or financial status and age.

Odds ratios should be presented with confidence intervals to allow for interpretation

Why divide AIC by N? (e.g. table 2)

Were models 1, 2, 3 done separately for grandmothers and grandfathers?

Hypothesis 1 as listed on line 302 differs from hypothesis 1 from the introduction. Furthermore, it is not clear that the model structure is directly addressing the original hypothesis (direct comparison between grandmothers and grandfathers).

Line 335 – your model structure means you should not state that Table 2 supports the hypothesis. The rest of the paragraph does provide support for this.

Line 420 – on the basis of the methods presented, your criticism here of this other work could equally apply to this paper.

Line 438 – did you control for multiple-testing here?

Reviewer #2: Thank you for this detailed exploration of the effect of grandparental care for grandchildren based on grandparents’ relationship status using the SHARE data. This is a detailed analysis that takes into consideration a number of variables that could potentially influence care provided by grandparents to grandchildren. The manuscript is well written and addresses the hypotheses well. There are no major issues that need revisions. The following are minor edits picked up during the review process. The areas for future research identified clearly and provides exciting pathways for other researchers to explore.

Lines 284-286 – the word “much” appears displaced in the sentence.

Line 302 – Consider rewording hypothesis 1 to be consistent with the introduction. “in intact couples” sounds awkward.

Line 310 – Consider stating “controlling for …” rather than “The addition of controls”

6. PLOS authors have the option to publish the peer review history of their article (what does this mean?). If published, this will include your full peer review and any attached files.

Reviewer #1: No

Reviewer #2: No

---

## [Author Response · Author response to Decision Letter 0]

3 Feb 2021

All of our response to the reviewer and editor comments are outlined, point-by-point, in the "Response to Reviewers" document already uploaded. They are not repeated here to avoid redundancy.

---

## [Editor Report · Decision Letter 1]

10 Feb 2021

PONE-D-20-38406R1

Grandparental partnership status and its effects on caring for grandchildren in Europe.

PLOS ONE

Dear Dr. Perry,

Thank you for submitting your manuscript to PLOS ONE. After careful consideration, we feel that it has merit but does not fully meet PLOS ONE’s publication criteria as it currently stands. Therefore, we invite you to submit a revised version of the manuscript that addresses the points raised during the review process.

Thank you very much for your thorough revision. Prior to final acceptance, I would like you to consider couple of more changes (below) that might make your results more transparent and strengthen your conclusions (and thus adhering to PLOS ONE publication criteria).

1.  Result and conclusions related to Table 1. It is my understanding that the omnibus F-tests were significant but it is unclear actually what groups differed significantly from each other as you do not provide more detailed analyses group-specific differences. Table 1 does not currently provide clear information to make such interpretations (naturally, since it is giving just descriptive statistics), so I would like you to give more information of statistical differences (probably corrected for multiple testing) to backup your conclusions. After all, you state in the Results that: "The results in Table 1 reinforce the justification for introducing each of these variables as controls". Moreover, since some of the responses used in Table 1 are percentages, the readers might wonder why logistic regression was not used in these comparisons.

2. To be consistent (e.g. with Table 2), please consider providing 95% CIs for odds ratios instead of asterisks indicating p-value-based significance in Table 4.

3. My previous minor point #3. To account for country, you added country as a fixed effect into the model. Was there some reason why you choose not to treat country as a random factor? To me, that would have made more sense statistically and would have resulted in less parameters fitted (1 vs. 19). 

We look forward to receiving your revised manuscript.

Kind regards,

Samuli Helle

Academic Editor

PLOS ONE

---

## [Author Response · Author response to Decision Letter 1]

7 Mar 2021

All of our responses are included in the "response to reviewers" that is already uploaded. To add it here would create unnecessary redundancy, but if it is required, please advise, and I will submit it here.

---

## [Editor Report · Decision Letter 2]

9 Mar 2021

Grandparental partnership status and its effects on caring for grandchildren in Europe.

PONE-D-20-38406R2

Dear Dr. Perry,

We’re pleased to inform you that your manuscript has been judged scientifically suitable for publication and will be formally accepted for publication once it meets all outstanding technical requirements.

Kind regards,

Samuli Helle

Academic Editor

PLOS ONE
---

## [Editor Report · Acceptance letter]

11 Mar 2021

PONE-D-20-38406R2 

Grandparental partnership status and its effects on caring for grandchildren in Europe. 

Dear Dr. Perry:

I'm pleased to inform you that your manuscript has been deemed suitable for publication in PLOS ONE. Congratulations! Your manuscript is now with our production department. 

Kind regards, 

on behalf of

Dr. Samuli Helle 

Academic Editor

PLOS ONE